# Abortive Zoonoses in Benin: Knowledge, Attitudes and Perceptions Gap Among Front-Line Small-Ruminant Production Stakeholders

**DOI:** 10.3390/ani15233405

**Published:** 2025-11-25

**Authors:** Ange-Régis Nonvignon Zoclanclounon, Camus Mahougnon Adoligbe, Bruno Enagnon Lokonon, Gloria Ivy Mensah, Benjamin Obukowho Emikpe, Souaïbou Farougou, Bassirou Bonfoh, Kennedy Kwasi Addo, Cyrille Kadoéïto Boko

**Affiliations:** 1Research Unit on Communicable Diseases (URMaT), University of Abomey-Calavi, Abomey-Calavi P.O. Box 01 BP 2009, Benin; 2Laboratoire de Biomathématiques et d’Estimations Forestières (LABEF), Faculty of Agronomic Sciences (FSA), University of Abomey-Calavi, Abomey-Calavi P.O. Box 01 BP 2009, Benin; brunolokonon@gmail.com; 3Centre Suisse de Recherches Scientifiques (CSRS), Abidjan P.O. Box 01 BP 1303, Côte d’Ivoire; bassirou.bonfoh@csrs.ci; 4Noguchi Memorial Institute for Medical Research, University of Ghana, Legon, Accra P.O. Box LG-581, Ghana; 5School of Veterinary Medicine, Kwame Nkrumah University of Science and Technology, Kumasi AK-731-0297, Ghana

**Keywords:** abortion, sheep, goat, One Health, modeling, West Africa

## Abstract

This study investigated the understanding and practices of people in Benin who work with sheep and goats regarding diseases that cause animal abortions and can spread to humans. We surveyed four key groups: farmers, butchers, meat inspectors, and para-veterinarians. Key Findings: (1) Knowledge was low: Overall, respondents correctly answered only about half of the knowledge questions. (2) Attitudes were risky: Majority (71%) reported risky practices, such as handling aborted materials without protection. (3) Perception: While most people saw these diseases as a threat to their animals, this concern did not lead to safer behavior. (4) Profession matters: Trained staff (like meat inspectors) knew more and had safer attitudes than farmers and butchers. (5) Awareness ≠ Action: Knowing about the diseases did not mean people felt personally at risk or changed their practices. (6) To protect both animal and human health, Benin needs targeted training programs for farmers and butchers, combined with wider animal vaccination efforts.

## 1. Introduction

Livestock production is a structural driver of economic growth in most low and middle-income countries. It accounts for roughly 40% of agricultural added value worldwide and still contributes about one-third of agricultural Gross Domestic Product (GDP) in sub-Saharan Africa [1]. Beyond the farm gate, downstream segments like feed, slaughter, processing, transport, retail, veterinary and financial services employ up to 1.3 billion people, nearly one in five individuals on the planet [2]. In West Africa’s ECOWAS (Economic Community of West African States) zone, transhumance pastoralism supports employment, milk production and rangeland functions across borders, though partial protocol enforcement creates implementation challenges for cross-border disease control [3]. Small ruminant value chains demonstrate competitive market structures with marketing efficiencies approaching 79%, indicating significant economic opportunities for productivity improvements [4]. Livestock is therefore widely recognized as a pathway to inclusive growth, poverty reduction and resilience [5].

However, reproductive disorders caused by infectious agents increasingly threaten both productivity and public health in small-ruminant systems. Abortive diseases such as brucellosis (*Brucella melitensis*), Q fever (*Coxiella burnetii*), toxoplasmosis (*Toxoplasma gondii*) and chlamydiosis (*Chlamydia abortus*) are major but under-diagnosed in many parts of Africa [6]. Recent serological surveys across sub-Saharan Africa report varied seroprevalences: brucellosis 1.23–10.7% in small ruminants, Q fever up to 64.7% in aborted goats, toxoplasmosis 6.8–8.6%, and *Chlamydia abortus* 2.0–7.79% [6,7,8,9]. These infections cause abortion, stillbirth, weak offspring and infertility, leading to significant economic losses to farmers and posing zoonotic risks to humans with occupational exposure resulting in human seroprevalence of 2.6% in pastoral communities [9].

In humans, brucellosis typically presents as prolonged fever, joint pain and malaise, whereas toxoplasmosis and Q fever often result in mild, non-specific febrile illness that may be confused with other endemic infections such as malaria in diagnostic-limited settings [10,11]. Such misdiagnosis contributes to the underestimation of their true burden.

The resulting economic burden is considerable. Correctly diagnosed brucellosis treatment ranges from € 9 per patient in Tanzania to € 650 in Algeria [12]. Beyond direct medical costs, socio-economic impacts include reduced offspring production, decreased milk yields, and compromised sales revenues, disproportionately affecting smallholder livelihoods [13].

In Benin, livestock contributes 5.7% of national GDP and 14.9% of agricultural GDP; annual meat production exceeds 70,000 t [14]. Small ruminants represent about 60% of the national herd and provide over 12% of total dietary protein [14]. They support the livelihoods of more than 500,000 households as an equally large population at risk of neglected zoonoses [14]. Recent market analyses revealed that the purchase price of kid or lamb in Benin for breeding varied from 5000 XOF (8.13 dollars USD) to 10,000 XOF (16.26 dollars USD), depending on the location [4]. Benin’s first documented seroprevalence study of *Chlamydia abortus* in small ruminants found seropositivity rates of 7.79% and 6.49% in two southern poles, confirming enzootic ovine abortion as a local animal health and public health concern [8].

Risk factors for these diseases include poor hygiene during parturition, communal grazing, mixed-species husbandry and limited access to veterinary services [9]. Field studies identified specific risk determinants including history of abortion, parity, species mixing in herds, assisting births with bare hands, handling placental membranes without protection, transhumance movements, and informal slaughter practices [6,9,15]. However, local evidence on how these management conditions interact with human behavior and perception to influence disease persistence is still lacking.

To date, no study in Benin has examined how different professional groups within the small-ruminant value chain (farmers, butchers, meat inspectors, and para-veterinary agents) understand and respond to abortion-causing zoonoses. Existing studies in the region have largely focused on seroprevalence or clinical diagnosis, with limited attention to behavioral or awareness-related determinants. Recent knowledge, attitudes, and practices (KAP) studies across Africa reveal critical gaps: in Cameroon, small ruminant farmers had very low mean knowledge scores (0.10 ± 0.20) and risk-perception scores (0.12 ± 0.33) regarding abortive zoonoses [16]; in Zimbabwe, only 34% of farmers could correctly identify abortion causes and specific pathogen awareness was low (10% for *Brucella*, 6% for *C. abortus*, 4% for *T. gondii*) [6]; in Ethiopia, 54.2% of respondents handled placentas and aborted fetuses with bare hands, and only 5.8% achieved good knowledge scores [15]. These studies consistently document weak translation of knowledge into safer practices, with attitudes and practices often misaligned with knowledge levels [17]. This represents a critical knowledge gap, as effective prevention and reporting depend on the attitudes and practices of those in daily contact with animals and their products.

To address this gap, we conducted a cross-sectional study to document farmers’ knowledge, attitudes and practices related to abortion-causing zoonoses, and to identify management behaviors that may contribute to their persistence.

We hypothesized that (i) Knowledge, Attitudes and Perception (KAP) levels would vary significantly between professional groups, (ii) higher knowledge would be associated with more desirable attitudes toward biosecurity and hygiene, and (iii) socio-demographic and occupational factors would influence perception of zoonotic risk.

The study aimed to provide behavioral evidence needed to design feasible hygiene-based interventions targeting brucellosis and related pathogens in Benin’s small-ruminant sector.

## 2. Materials and Methods

### 2.1. Study Area

The study was conducted in the Republic of Benin (West Africa) and covered four communes selected from South to North: Parakou, Dassa-Zoumè, Cotonou and Abomey-Calavi (Figure 1).

Parakou (North), with a population of approximately 255,000 inhabitants, is a major commercial hub connecting northern livestock-producing areas with southern consumption markets [18,19]. The commune’s economy depends heavily on livestock trading, with several slaughterhouses and informal killing points representing potential hotspots for *Brucella* transmission to workers and consumers. Its tertiary-level health facilities also enable the clinical detection of human brucellosis.

Dassa-Zoumè (Central), a largely agrarian commune of about 120,000 inhabitants, is characterized by mixed crop–livestock systems, small family-run goat and sheep farms, and close daily contact between humans and animals, which increases zoonotic transmission risks [18,20].

Abomey-Calavi (South), with over 655,000 residents, is a peri-urban area where intensive animal keeping is common in backyards and smallholdings [18].

Cotonou (South), Benin’s economic capital and largest urban center (≈700,000 inhabitants), hosts the national wholesale abattoir that receives animals from all agro-ecological zones, making it a critical node for both economic activity and potential disease dissemination along the value chain [18,21].

These communes were selected to represent contrasting production systems, population densities, and animal movement patterns that influence exposure to abortive zoonoses.

### 2.2. Sampling

The study enrolled four professional groups that handle small ruminants or their products in the selected communes: (i) flock owners, (ii) butchers, (iii) veterinary or para-veterinary practitioners and (iv) meat-inspection officers. All participants were interviewed face-to-face using a structured digital questionnaire (KoBoCollect). The instrument recorded identification, education level, workplace characteristics, practical experience, and Knowledge, Attitude and Perception (KAP) regarding abortive zoonoses (Table 1 and Appendix A).

Prior to the survey, the questionnaire was pre-tested with ten respondents (two from each professional group) in a non-study commune to ensure clarity, relevance, and local comprehensibility; minor wording adjustments were made based on feedback.

For farmers, the sample size per commune was calculated with the normal approximation of the binomial:$$n=U_{1-\propto /2}^{2}\left[p\left(1-p\right)/d^{2}\right]$$

With n the number of farmers to be interviewed. We used $U_{1-\propto /2}=1.96$ for ∝=0.05. p represented the estimated proportion of small-ruminant farmers per commune in Benin, and d=5% was the allowable margin of error for all study estimates.

According to the National Agricultural Register [14], Benin had a total of 577,703 sheep and goat keepers. An assumed equitable distribution of keepers among the country’s 77 districts gives a probability of *p* = 1.30% keepers per district. The resulting sample size using the formula was 19.7, rounded to 20 farmers per commune.

Veterinary and para-veterinary officers were selected based on the official lists provided by local veterinary services, ensuring at least 20 individuals per commune for comparison with farmers.

For butchers and meat inspectors, the target was to interview all available personnel working with small ruminants in the main slaughter or retail facilities of each commune during the study period and, respectively, 22 and 7 of them were interviewed.

### 2.3. Data Analysis

Data captured was exported to Microsoft Excel (version 16.100.3) as XML, and cleaned to remove duplicates or inconsistencies. Analyses were run in R 4.1.1. Maps were drawn with QGIS version 3.38.3-Grenoble.

#### 2.3.1. KAP Levels Around Respondents

Descriptive statistics (mean, SD, counts) summarized socio-demographics and item responses. Raw correct and desirable answers rate were determined. For the attitude scale, options 1 (strongly disagree) and 2 (disagree) were collapsed into “desirable”, 3 (agree) and 4 (strongly agree) into “undesirable. The coding was reversed for perception items (1–2 are “undesirable” and 3–4 “desirable”).

To assess the factors associated with the participants’ knowledge, attitude, and perception scores toward zoonotic diseases, a univariable mixed-effects linear regression was fitted. Each categorical predictor was included in the model with a designated fixed (reference) category, against which the other categories were compared. Specifically, *female*, *butcher*, *rural location*, *no multisite work*, *no contact with animals*, 0 h *of contact duration*, *no animal abortion observed*, and *no human symptoms reported* were defined as the reference groups.

The regression coefficients therefore represent the mean difference in knowledge, attitude, or perception scores for each category relative to its respective reference. Statistical significance was assessed using *p*-values (<0.05), with estimates indicating the direction and magnitude of associations between participant characteristics and outcome scores.

#### 2.3.2. KAP Correlations

Relationships between sub-scales were evaluated with Pearson’s r (|r| ≥ 0.80 very strong; 0.60–0.79 strong; 0.40–0.59 moderate; 0.20–0.39 weak; <0.20 negligible).

#### 2.3.3. Factor Influencing Zoonotic Risks

Each unidimensional sub-scale was finally fitted to a two-parameter logistic IRT model (2-PL) using marginal maximum likelihood.$$Pij(uij=1|\theta j)=\frac{1}{1+e^{-1,7\;a_{i}(\theta j-b_{i})}}$$
where *a_i_* denoted item discrimination (ability of the item to differentiate respondents with differing latent traits) and *b_i_* item difficulty (trait level required for a 50% chance of a correct/desirable response).

Because the total sample size was relatively small (<200 participants), a non-parametric bootstrap with 1000 replications approach was additionally applied to obtain more robust estimates and 95% confidence intervals for both discrimination and difficulty parameters. This resampling-based correction helped reduce potential bias and improve parameter stability under limited sample conditions.

Questions with ≤0.70 or with an almost flat ICC between θ=−4 and +4 were removed, as such items fail to differentiate between low- and high-scoring respondents and therefore weaken construct validity [22].

## 3. Results

### 3.1. Respondent Profile

The data summary shown that male respondents were more represented (82%) compared with females (18%). Farmers constituted the largest professional group (50%), followed by para-veterinary technicians (25%) and butchers (19%). With regard to professional experience, 67% of participants had been active for less than 10 years.

### 3.2. Working Conditions and Attitudes

Table 2 describes the respondents’ working environments and routines. Meat inspectors and para-veterinarians reported the longest daily contact with animals. Approximately 96% of para-veterinarians and 75% of meat inspectors moved between several workplaces. All inspectors and para-veterinarians, as well as about 88% of farmers and 68% of butchers, declared direct physical contact with animals. The mean duration of contact was less than 1 h for 23% of respondents, a category largely represented by farmers. Long exposures (more than 5 h per day) were most common among meat inspectors (42.9%) and butchers (31.8%).

### 3.3. Abortion Events in Small Ruminants

Table 3 shows data on caprine and ovine abortions. More than 30% of respondents had encountered at least one case of abortion observations of abortion cases in animals under their care. Dassa-Zoumè (50%) and Abomey-Calavi (42.3%) recorded the highest proportions. In most cases (15.7%), respondents had observed a single abortion episode. Abortions were reported throughout the year regardless of season by 84.35% of interviewees. Goats were affected more frequently (37.14%) than sheep (28.57%).

### 3.4. KAP Analysis

#### 3.4.1. KAP Levels Around Respondents

Para-veterinarians and meat inspectors demonstrated the highest levels of knowledge, consistent with their formal training and roles within the veterinary public health system. Butchers, conversely, consistently presented the lowest scores across all three domains, indicating a critical gap in understanding, the prevalence of risky practices, and a lower perception of personal risk.

Farmers exhibited a notable profile while their knowledge was moderate and their attitudes were moderately desirable; they displayed the highest perception of risk among all groups. The collective profile of butchers characterized by low knowledge, the least desirable attitudes, and a lower sense of personal vulnerability identifies them as a particularly high-risk group and a priority target for tailored intervention programs (Figure 2).

The univariable mixed-effects linear regression (Section A.1) revealed distinct patterns across sub-scales. Knowledge was significantly higher among meat inspectors (+0.49, *p* < 0.001), para-vets (+0.38, *p* < 0.001), urban or multi-site workers (+0.18 each, *p* < 0.001) and respondents who had experienced an abortion (+0.17, *p* = 0.001), whereas sex had no effect.

Conversely, riskier attitude scores decreased for para-vets (−0.31, *p* < 0.001), inspectors (−0.24, *p* = 0.005), urban (−0.14, *p* = 0.001) and multi-site employment (−0.23, *p* < 0.001) as well as after an abortion episode (−0.14, *p* = 0.003); a contact duration of 1 to 3 h also lowered the score (−0.23, *p* = 0.019). Regarding profession and taking butchers as reference, farmers (+0.27), inspectors (+0.37) and para-vets (+0.25) reported a higher desirable perceived risk (all *p* < 0.001). Location, multi-site work and animal contact were non-significant, but self-reported zoonotic symptoms coincided with a lower desirable perception (−0.17, *p* = 0.024).

The internal item reliability test (Cronbach’s test) shown, after removal of the invariant item K6.1 and reverse-coding of AT1, AT6, P7 and P8, Cronbach’s α reached 0.85 for knowledge and 0.745 for attitudes, and 0.67 for perception below the 0.70 reliability benchmark (Section A.2).

#### 3.4.2. KAP Correlations

Pearson correlations (Section A.3) showed a moderate, inverse relationship between correct knowledge and risky attitude (r = −0.56, 95% CI −0.68 to −0.42; *p* < 0.001; r^2^ ≈ 0.31). The link between knowledge and perception was weak and non-significant (r = 0.14; *p* = 0.135), and attitude was not associated with perception (r = −0.05; *p* = 0.583).

#### 3.4.3. Factor Influencing Zoonotic Risks

Figure 3 and Section A.4 illustrate the item characteristic curves (2-PL) obtained after recalibrating the 28 retained items. Each panel plotted the probability P(uij=1∣θ) of a correct response (or desired attitude/perception) against latent ability θ.

Discrimination values ranged from −3.07 (AT3) to 7.23 (K5), indicating large differences in the ability of items to distinguish between respondents with low and high levels of the underlying trait. Items such as K5 (7.23) and K6.3 (5.92) showed excellent discrimination among professional groups. Conversely, several attitude-related items (AT2–AT7) exhibited negative discrimination values.

Regarding item difficulty, estimates varied widely from very low difficulty to give a correct or desirable answer (AT8 = −30.95, P1 = −3.01) to very high ones (K7 = 11.60, K4.3 = 3.40).

IRT re-fit after removing flat curves confirmed the robustness of the Knowledge sub-scale (Section A.5).

The eight retained items displayed a high mean discrimination (a = 6.2) and difficulties from b = −1.1 to b = 2.0. “Name at least three abortive diseases” (K4.3) was the best performer (a ≥ 10), (b ≈ 2), finely separating well-informed respondents, whereas the basic “animal-to-human transmission” item (K1) was easy (b ≈ −1) but still discriminating (a ≈ 2.4).

The Perception sub-scale showed moderate discrimination (a ≈ 0.9) and clearly negative difficulties, indicating that most participants readily recognized the severity of abortive diseases. Items on danger to animals or humans (P1, P2) were almost always endorsed (b ≤ −3). The statement “I already take enough precautions” (P8) demanded higher ability (b ≈ −0.9) and was the most informative perception item (a ≈ 1.5).

By contrast, the Attitude block retained only one useful question after filtering. Despite its extreme ease (b ≈ −14), it discriminated poorly (a ≈ 0.1), revealing a pronounced ceiling effect: virtually all respondents claimed they protected themselves when handling an aborted fetus.

Bootstrap resampling (Section A.6) indicates considerable variability in the precision of item parameter estimates across the full set of items. Items such as K5 (a = 10.668, CI [3.991, 37.763]) and K6.3 (a = 10.231, CI [3.155, 40.632]) again displayed very high discrimination values, consistent with their strong ability to differentiate respondents, although the wide confidence intervals suggest a degree of uncertainty or instability in these estimates. Similarly, items K2 (a = 5.135, CI [1.154, 30.533]) and K3 (a = 3.785, CI [1.803, 9.477]) showed moderately high discrimination with narrower CIs, indicating more stable performance.

Several “K” items (notably K4.1, K4.2, and K6.2) also demonstrated satisfactory discrimination (ranging from 1.5 to 3) with relatively tight confidence intervals, supporting their reliability as part of the measurement model. In contrast, many “AT” and “P” items presented low or negative mean discrimination values, such as AT3 (a = −3.665, CI [−8.123, −1.838]), AT7 (a = −2.590, CI [−4.557, −1.271]), and P5 (a = −0.607, CI [−1.212, −0.014]), reflecting weaker differentiation between high- and low-ability respondents.

Regarding the difficulty parameters (b), there was a wide dispersion across items, with mean values ranging from very low (e.g., AT1 = −5.237, P1 = −3.120) to very high (K7 = 6.658, P3 = 5.793). Some items, such as AT6, AT8, and P2, exhibited extremely large standard deviations and implausibly wide confidence intervals spanning several orders of magnitude indicating that their difficulty estimates were highly unstable under resampling.

## 4. Discussion

This study documented critical gaps in knowledge, attitudes, and perceptions regarding abortive zoonoses among four professional groups within Benin’s small-ruminant value chain. Our findings reveal low pathogen-specific awareness, persistent risky practices, and weak translation of knowledge into protective behaviors. These patterns reflect broader structural constraints in veterinary service delivery, gendered information flows, and economic incentives that shape livestock disease management across West Africa.

### 4.1. Knowledge Gaps and Their Structural Determinants

Producers, butchers, meat inspectors, and para-veterinarians showed limited knowledge of abortive pathogens, particularly regarding clinical signs (K6) and zoonotic transmission routes. Only one-fifth of respondents recognized placental contact as hazardous, and more than two-thirds disposed of aborted material bare-handed. Similar deficits have been documented across sub-Saharan Africa. In Kenya, only 12% of farmers understood animal-to-human transmission pathways [23]. In Zimbabwe, specific awareness of *Brucella* (10%), *C. abortus* (6%), and *T. gondii* (4%) was equally low despite 44% reporting recent reproductive problems [6]. In Ethiopia, 54.2% of respondents handled placentas with bare hands, and only 5.8% achieved good knowledge scores [15].

These persistent gaps are not simply individual ignorance but reflect systemic determinants. Limited veterinary surveillance and diagnostic capacity reduce awareness and control prioritization at national levels across West Africa [24]. Educational attainment strongly predicts safer practices; in Tanzania, farmers with tertiary education were significantly less likely to self-administer antimicrobials (76.6% of respondents reported self-treatment). In north-west Côte d’Ivoire, veterinary messaging predominantly targeted men, leaving women who perform most dairy processing with lower access to prevention information despite their higher exposure risk [25]. Economic constraints also shape behavior. A six-country West African study (n = 728 peri-urban dairy farms) found marked heterogeneity in commercialization levels, which affected farmers’ willingness to adopt vaccination and biosecurity measures [26].

Our regression analyses confirmed these patterns. Knowledge scores were significantly higher among meat inspectors and para-veterinarians, reflecting formal training advantages. Urban and multi-site workers also scored better, likely due to greater exposure to veterinary services and information networks. Respondents who had directly experienced abortions showed higher knowledge, suggesting that experiential learning occurs but only after economic losses.

### 4.2. Abortion Patterns and Endemic Transmission

Abortions occurred year-round rather than seasonally in our study communes. This pattern suggests endemic transmission driven by persistent management failures rather than climatic peaks. It contrasts with rainfall-linked seasonality observed in Senegal [27] and implies that continuous circulation of *Brucella* and allied pathogens requires non-seasonal, sustained interventions. Year-round occurrence also reflects the absence of synchronized breeding or calving management in Benin’s extensive small-ruminant systems.

Transhumance and livestock mobility amplify transmission risk. In Mali, individual *Brucella* seroprevalence reached 8.2% (herd-level 21.2%), with mobility status and abortion history significantly associated with infection [28]. Benin’s cross-border pastoral movements and communal grazing practices likely sustain similar transmission dynamics, though our study did not collect mobility data. Integrating market and border testing into surveillance systems has been recommended to address mobile herd risks [28].

### 4.3. Occupational Exposure and Under-Recognition of Human Cases

Butchers and para-veterinarians reported the highest prevalence of self-reported “brucellosis-like symptoms” (prolonged fever, joint pain). However, this finding must be interpreted cautiously, our study did not collect laboratory confirmation, and symptom overlap with malaria and other febrile illnesses. In a Nigerian study only 10% of respondents had ever suspected brucellosis, far below the 30% attribution rate for unexplained fevers found in Nigeria [29]. This under-recognition likely stems from weak clinical awareness, limited laboratory access, and the non-specific nature of brucellosis presentation. Occupational seroprevalence studies in pastoral communities have documented human seropositivity rates of 2.6% [9], confirming zoonotic transmission but highlighting the need for integrated human–animal surveillance to capture true burden.

### 4.4. Knowledge-Attitude-Perception Relationships

Our correlation analyses revealed a moderate inverse relationship between knowledge and risky attitudes. This suggests that better information fosters safer behavior, consistent with findings from Saudi Arabia and Bangladesh where formal training improved KAP metrics [30,31]. However, knowledge was not significantly linked to risk perception, and attitude was unrelated to perception. This dissociation indicates that factual awareness alone does not translate into a sense of personal vulnerability.

These findings underscore the need for risk communication strategies that go beyond information transfer. Behavioral interventions combining practical demonstrations, visual aids, and participatory learning have been implemented in African dairy and pastoral settings [26].

### 4.5. Psychometric Performance and Scale Validation

Our IRT analyses provided nuanced insights into item performance. The Knowledge sub-scale demonstrated strong psychometric properties after refinement. Eight retained items displayed high mean discrimination and difficulties ranging from −1.1 to 2.0. “Name at least three abortive diseases” (K4.3) was the best performer, effectively separating well-informed respondents. Conversely, basic items like “animal-to-human transmission” (K1) were easy but still discriminated adequately.

The Perception sub-scale showed moderate discrimination and clearly negative difficulties, indicating that most participants readily recognized the severity of abortive diseases. Items on danger to animals or humans (P1, P2) were almost always endorsed creating a ceiling effect. The most informative perception item was “I already take enough precautions”.

The Attitude sub-scale posed greater challenges. Several items exhibited negative discrimination values, suggesting misinterpretation, poor wording, or reverse-coding issues. After filtering, only one attitude item remained useful, and it discriminated poorly despite extreme ease. This ceiling effect virtually all respondents claimed they protected themselves when handling aborted fetuses likely reflects social desirability bias, a common limitation in self-reported KAP data [32,33].

Bootstrap resampling confirmed considerable variability in parameter estimates. Items K5 and K6.3 displayed very high discrimination but wide confidence intervals, indicating instability due to our modest sample size (n < 200). This instability is not uncommon in small-sample IRT applications and underscores the value of non-parametric resampling to characterize uncertainty [34].

Internal consistency varied across sub-scales and the perception fell below the conventional 0.70 threshold. Lower alpha can occur in multidimensional constructs or short scales and does not necessarily invalidate the measure [35]. Recent KAP validation studies have accepted α ≈ 0.6 for newly developed scales, particularly when complemented by item-total correlations and IRT fit indices [36,37]. Our iterative approach combining 2-PL IRT, Cronbach’s α, and expert review aligns with best practices documented in leptospirosis, chronic kidney disease, and confined-space KAP validations [34,35,37].

### 4.6. One Health Policy Implications

Brucellosis and other abortive agents (*Coxiella burnetii, Chlamydia abortus, Toxoplasma gondii*) require integrated animal–human strategies. West African policy frameworks increasingly emphasize One Health approaches, but implementation gaps persist. Laboratory and diagnostic services remain constrained, limiting confirmation of zoonotic infections and undermining surveillance [24].

Our results suggest two priority interventions for Benin. First, scale up flock vaccination using context-appropriate delivery models. A six-country West/Central African study proposed public–private partnership (PPP) models tailored to local commercialization levels, arguing that transactional delivery (the most common current model) is suboptimal in many settings [26]. Sustained vaccination has markedly reduced bovine and human brucellosis, where implemented and should be combined with supervised disposal of fetal products (P5 item) to address the high frequency of unprotected handling (AT9 item).

Second, deliver context-specific training that combines practical demonstrations with visual aids, targeting farmers and butchers who lack formal biosafety instruction. Training should be gender-sensitive, ensuring women involved in dairy processing and flock management receive tailored messages [25]. Integrating market and border testing for transhumant herds can address mobility-related risks [28]. Field Epidemiology and Laboratory Training Programs (FELTP) that include veterinary epidemiology have strengthened outbreak detection and One Health responses in Nigeria and represent scalable capacity investments [38].

## 5. Conclusions

Actors along Benin’s small-ruminant chain possessed inconsistent knowledge of abortive zoonoses; with poor recognition of key pathogens, transmission pathways, and clinical signs. This deficit, coupled with entrenched high-risk behaviors, sustains pathogen circulation and exposes both livestock and humans to preventable infection. Targeted One Health interventions education, vaccination, and better working conditions are vital to disrupt the cycle and protect public health.

## Figures and Tables

**Figure 1 animals-15-03405-f001:**
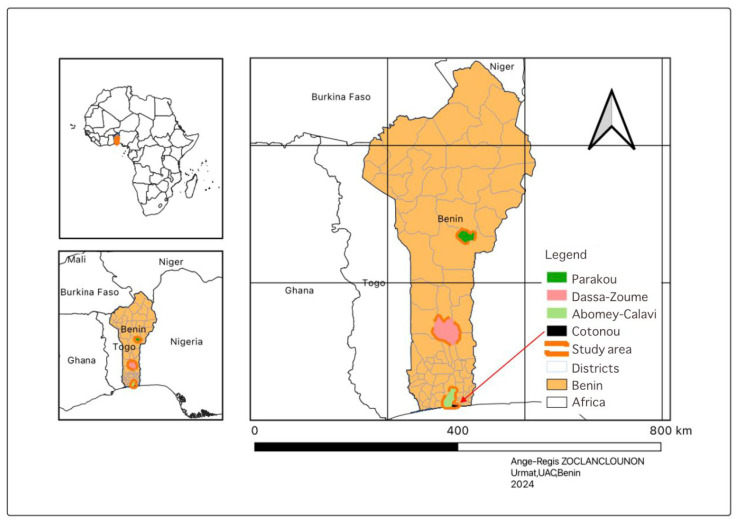
Study Area.

**Figure 2 animals-15-03405-f002:**
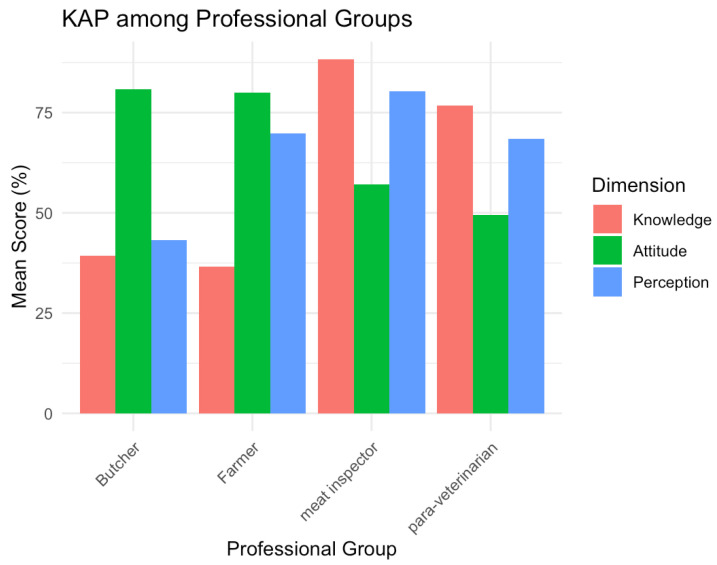
KAP Among Professional Groups.

**Figure 3 animals-15-03405-f003:**
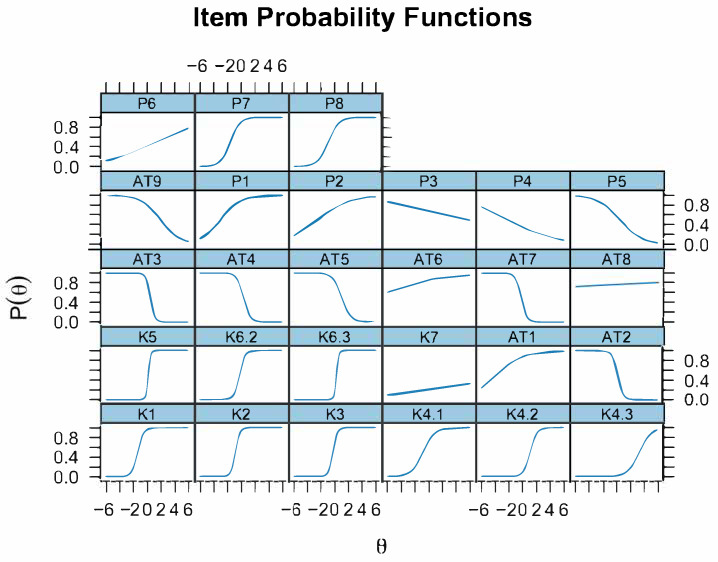
2-PL Characteristics.

**Table 1 animals-15-03405-t001:** KAP questions addressed to respondents.

Code	Sub-Scale	Question to Ask
K1	Knowledge	Do you know that an animal can become sick even when no wound is visible?
K2	Knowledge	Do you think that a disease in an animal can also make a human sick?
K3	Knowledge	Have you ever heard of diseases that cause abortion in goats or sheep?
K4.1	Knowledge	Can you name one disease that causes females to abort?
K4.2	Knowledge	Can you name two different diseases that cause abortion?
K4.3	Knowledge	Can you name three or more such diseases?
K5	Knowledge	Have you ever heard the term “brucellosis”?
K6.1	Knowledge	Give one sign observed in an animal with brucellosis.
K6.2	Knowledge	Give two signs of brucellosis.
K6.3	Knowledge	Give three or more signs of brucellosis.
K7	Knowledge	Which species can transmit these diseases? (goat/sheep/cattle/dog)
AT1	Attitude	In the event of an abortion, I handle the fetus with bare hands.
AT2	Attitude	I do not isolate the aborting female from the flock.
AT3	Attitude	Cleaning or disinfecting the ground after an abortion is unnecessary.
AT4	Attitude	After an abortion I do not treat the female.
AT5	Attitude	I never call a veterinarian for an abortion case.
AT6	Attitude	Herbal remedies are always sufficient to treat an aborting female.
AT7	Attitude	I let the female recover on her own without intervention.
AT8	Attitude	I quickly sell the female that has aborted.
AT9	Attitude	I do not vaccinate my animals against these diseases.
P1	Perception	Abortive diseases are a serious danger for the health of my animals.
P2	Perception	These diseases are also a danger for my own health and that of my family.
P3	Perception	Abortions can severely slow the development of my herd.
P4	Perception	Culling females after an abortion is a solution.
P5	Perception	Vaccination is useful to prevent these diseases.
P6	Perception	Abortive diseases are not really a problem in my area.
P7	Perception	My occupation increases my risk of contracting an abortive disease.
P8	Perception	I do not take enough precautions to avoid being infected.

**Table 2 animals-15-03405-t002:** Working conditions and attitudes.

Variable	Butcher (N = 22)	Farmer (N = 57)	Meat Inspector (N = 7)	Para-Vet (N = 29)	Total (N = 115)
Multisite work					
*No*	2 (20%)	46 (81%)	1 (25%)	1 (3.8%)	50 (43%)
*Yes*	8 (80%)	11 (19%)	3 (75%)	25 (96%)	47 (41%)
*No response*	12	0	3	3	18 (16%)
Direct contact with animals					
*No*	0 (0%)	7 (12%)	0 (0%)	0 (0%)	7 (6%)
*Yes*	15 (100%)	50 (88%)	7 (100%)	27 (100%)	99 (86%)
*No response*	7	0	0	2	9 (8%)
Contact duration					
0 h	0 (0%)	6 (10.5%)	0 (0%)	0 (0%)	6 (5%)
< 1 h	1 (4.5%)	24 (42.1%)	1 (14.3%)	0 (0%)	26 (23%)
1–3 h	2 (9.1%)	13 (22.8%)	0 (0%)	1 (3.4%)	16 (14%)
3–5 h	2 (9.1%)	6 (10.5%)	0 (0%)	0 (0%)	8 (7%)
> 5 h	7 (31.8%)	1 (1.8%)	3 (42.9%)	2 (6.9%)	13 (11%)
*Unknown*	10 (45.5%)	7 (12.3%)	3 (42.9%)	26 (89.7%)	46 (40%)

N = number of respondents.

**Table 3 animals-15-03405-t003:** Animal abortion events.

	No	Yes	Respondents
Commune			
*Abomey-Calavi*	15 (18.75%)	11 (31.43%)	26 (22.61%)
*Cotonou*	8 (10.00%)	0 (0.00%)	8 (6.96%)
*Dassa-Zoumè*	8 (10.00%)	8 (22.86%)	16 (13.91%)
*Parakou*	49 (61.25%)	16 (45.71%)	65 (56.52%)
**Number of abortions**			
*1 time*	2 (2.50%)	16 (45.71%)	18 (15.65%)
1–3 *times*	0 (0.00%)	9 (25.71%)	9 (7.83%)
*3*–6 *times*	0 (0.00%)	1 (2.86%)	1 (0.87%)
6+ *times*	0 (0.00%)	9 (25.71%)	9 (7.83%)
*NA*	78 (97.50%)	0 (0.00%)	78 (67.83%)
**Abortion season**			
*Both seasons*	78 (97.50%)	19 (54.29%)	97 (84.35%)
*Rainy*	2 (2.50%)	14 (40.00%)	16 (13.91%)
*Dry*	0 (0.00%)	2 (5.71%)	2 (1.74%)
**Species affected**			
*Goat*	0 (0%)	13 (37.14%)	13 (11.30%)
*Sheep*	0 (0%)	10 (28.57%)	10 (8.70%)
*Both*	80 (100%)	12 (34.29%)	92 (80.00%)
**Total**	**80 (69.57%)**	**35 (30.43%)**	**115 (100%)**

## Data Availability

The data that support the findings of this study are available from the corresponding author upon reasonable request.

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
