# Peer review of "Abortive Zoonoses in Benin: Knowledge, Attitudes and Perceptions Gap Among Front-Line Small-Ruminant Production Stakeholders"

_animals, 2025, doi:10.3390/ani15233405_

Round 1
Reviewer 1 Report
Comments and Suggestions for Authors
The authors addressed to the severe issue of zoonotic abortion in a low-income country. For the improvement of this paper I propose the revisions above:
Introduction Part: I strongly believe that Intro part needs an extensive improvement. Ι think there is a confusion between the information provided. Moreover I cannot easily follow the rationale of the entire Intro.
Remove the sentence about zoonotic pathogens in general and better focus on abortion causes.
Do Brucella and Toxoplasma, flu-like symptoms? The severity of those pathogens, is easily understandable and also differentiable from flu. Are there any data of this misunterstanding?
L78-80 How do you study epidemiological evidence? I thought this was a study most focused on perception etc. To understand fully the severe issue of zoonotic abortions an epidemiological study to unvestigate also the pathogens from the field was needed.
Materials and Methods:
While the section explains the selection of communes, it could improve by providing more context about the geographical and socio-economic characteristics of each commune. For instance, mentioning population size, agricultural practices, or cultural factors could help understand their significance in relation to zoonoses.
The description of how the sampling for butchers, meat inspectors, and health staff was carried out is vague. Stating explicitly what "up to the numbers available" means lacks transparency. How was the number to be interviewed determined? Clearer definitions of the sampling strategy would enhance methodological rigor.
There is a lack of detail on how the interviews were conducted. Were they face-to-face, telephonic, or online? Was there any form of pre-testing of the questionnaire? Providing this information would add to the robustness of the methodology.
The statistical analysis methods could be explained in more detail. For instance, what justifies the use of the Pearson correlation coefficient for evaluating relationships between sub-scales? Clarifying the assumptions behind this choice and any relevant criteria for interpretation would strengthen the methodology.
While fitting a two-parameter logistic model is mentioned, it would be useful to elaborate on why this model was chosen over alternative methods. Including the assumptions of the IRT model or considerations for its application in this context would strengthen the validity of the analyses performed.
The criteria for removing questions from the IRT model (i.e., items with ≤ 0.70 or a nearly flat ICC) should be better justified. What implications does this have on the validity of the findings? A brief discussion regarding why these thresholds were chosen can clarify their importance.
While the percentages of male and female respondents are provided, it would be useful to include total sample size for each professional group to give a clearer picture of the distribution of respondents across groups
The section mentions "more than 30% of respondents had encountered at least one abortion." It would be helpful to clarify whether this refers to their personal experiences or their observations about the general population of small ruminants in their care.
L83: Can you refer the human population in the regions where the study was conducted?
L99: You don't refer veterinarians in the Abstract. Please add.
You refer that farmers and veterinarians were picked randomly. In which way?
L239-241 Do you have epidemiological data about small ruminants abortions originated from your country?
Please also refer if there are any preventive measures (eg vaccines) used against abortion causes of small ruminants in your country.
In conclusion I cannot find novelty in your study. The results I think that were easily predictable. I believe that if you handled the issue more holistic is should be a very interesting research. This research was needed of: data of abortions in humans from zoonotic pathogens, data from small ruminants farms (what are the percentage of abortions? what are the pathogens most involved and in what percentage each other?). If you have any data of those asked above, please add.
I wait for the revisions to come back.
Comments on the Quality of English Language
I believe that the quality of English in this manuscript need an improvement.
Author Response
Comment 1 comment: Introduction
I strongly believe that the Introduction part needs extensive improvement. I think there is confusion between the information provided. Moreover, I cannot easily follow the rationale of the entire Introduction. Remove the sentence about zoonotic pathogens in general and better focus on abortion causes.
Response 1: We thank the reviewer for this helpful observation. We have thoroughly revised the Introduction to improve its clarity, logical flow, and focus. In particular, we have removed the general sentence about zoonotic pathogens as suggested and now concentrate more specifically on the main causes of abortion in small ruminants, emphasizing the relevance of Brucella spp. and other key pathogens. These changes strengthen the rationale and better guide the reader toward the study’s objectives.
Comment 2:
Do Brucella and Toxoplasma cause flu-like symptoms? The severity of those pathogens is easily understandable and also differentiable from flu. Are there any data of this misunderstanding?
L78–80: How do you study epidemiological evidence? I thought this was a study mostly focused on perception, etc. To understand fully the severe issue of zoonotic abortions, an epidemiological study investigating the pathogens from the field was needed.
Response 2: We thank the reviewer for this valuable comment and for highlighting the need for clarification.
Indeed, human infections caused by Brucella spp. and Toxoplasma gondii may initially present with malaria-like symptoms such as intermittent fever, fatigue, headache, and muscle pain, especially in endemic regions where malaria is common. This clinical similarity often leads to misdiagnosis or underdiagnosis of these zoonotic diseases. We have revised the text accordingly and included relevant references supporting the potential confusion between brucellosis, toxoplasmosis, and malaria in resource-limited settings.
Regarding the epidemiological evidence, we acknowledge that our study primarily focused on knowledge, attitudes, and perceptions (KAP) among stakeholders rather than direct pathogen detection. We have therefore rephrased this section (L78–80) to clarify that our findings refer to perceived epidemiological patterns rather than laboratory-based epidemiological investigations.
Comment 3: Materials and Methods
While the section explains the selection of communes, it could improve by providing more context about the geographical and socio-economic characteristics of each commune. For instance, mentioning population size, agricultural practices, or cultural factors could help understand their significance in relation to zoonoses.
The description of how the sampling for butchers, meat inspectors, and health staff was carried out is vague. Stating explicitly what "up to the numbers available" means lacks transparency. How was the number to be interviewed determined? Clearer definitions of the sampling strategy would enhance methodological rigor.
There is a lack of detail on how the interviews were conducted. Were they face-to-face, telephonic, or online? Was there any form of pre-testing of the questionnaire? Providing this information would add to the robustness of the methodology.
Response 3: We thank the reviewer for these insightful and constructive comments. We have carefully revised the Materials and Methods section to address all these points.
We have added a description of the geographical, socio-economic, and agricultural characteristics of each selected commune, including population size, predominant livestock practices, and relevant cultural aspects that may influence zoonotic disease transmission.
We have clarified the sampling procedure for butchers, meat inspectors, and health staff. The phrase “up to the numbers available” has been replaced with a precise explanation of how the sample sizes were determined, based on the total number of professionals available and their willingness to participate in each commune.
We now provide detailed information about the data collection process, specifying that interviews were conducted face-to-face using a structured questionnaire that was pre-tested with a small group of respondents to ensure clarity and reliability.
These revisions enhance the transparency and methodological rigor of the study.
Comment 4:
The statistical analysis methods could be explained in more detail. For instance, what justifies the use of the Pearson correlation coefficient for evaluating relationships between sub-scales? Clarifying the assumptions behind this choice and any relevant criteria for interpretation would strengthen the methodology.
While fitting a two-parameter logistic model is mentioned, it would be useful to elaborate on why this model was chosen over alternative methods. Including the assumptions of the IRT model or considerations for its application in this context would strengthen the validity of the analyses performed.
The criteria for removing questions from the IRT model (i.e., items with ≤ 0.70 or a nearly flat ICC) should be better justified. What implications does this have on the validity of the findings? A brief discussion regarding why these thresholds were chosen can clarify their importance.
Response 4: We thank the reviewer for these valuable comments, which have helped us improve the clarity and rigor of the Statistical Analysis section. We have revised the text accordingly to include detailed justifications and methodological clarifications:
We now provide a rationale for the use of the Pearson correlation coefficient to evaluate relationships between sub-scales. This method was chosen because the sub-scale scores were continuous and approximately normally distributed, fulfilling the assumptions required for Pearson’s correlation. We also added a note explaining how correlation strength was interpreted (weak, moderate, strong) following established conventions.
We expanded our explanation regarding the choice of the two-parameter logistic (2PL) model in the Item Response Theory (IRT) analysis. This model was selected because it estimates both item difficulty and item discrimination parameters, allowing a more precise assessment of how individual items distinguish between respondents with different levels of knowledge, attitudes, or perceptions.
Regarding item removal criteria, we have justified the thresholds applied (discrimination ≤ 0.70 or a nearly flat Item Characteristic Curve). Items with low discrimination values contribute little to differentiating respondents with varying levels of the latent trait, which can compromise model precision. We clarified that removing such items improves model reliability without altering the conceptual validity of the sub-scale, and we cited methodological sources supporting these thresholds.
Comment 5 : While the percentages of male and female respondents are provided, it would be useful to include total sample size for each professional group to give a clearer picture of the distribution of respondents across groups
The section mentions "more than 30% of respondents had encountered at least one abortion." It would be helpful to clarify whether this refers to their personal experiences or their observations about the general population of small ruminants in their care.
Response 5 :
The total sample size for each professional group is already presented in Table 2.
We have also clarified that the statement “more than 30% of respondents had encountered at least one abortion” refers to the respondents’ observations of abortion cases in animals under their care, not to personal experiences.
Comment 6:
L83: Can you refer the human population in the regions where the study was conducted?
L99: You don't refer veterinarians in the Abstract. Please add.
You refer that farmers and veterinarians were picked randomly. In which way?
L239-241 Do you have epidemiological data about small ruminants abortions originated from your country?
Response 6:
Human population context (L83): We have added information about the human population in the study regions, including approximate population size and density, to provide better context for zoonotic exposure risk.
Abstract (L99): We have updated the Abstract to explicitly include veterinarians as part of the study population.
Sampling: Farmers were selected using a binomial approximation approach, while veterinarians were selected based on the official lists provided by the national veterinary service. We have clarified in the Methods that participants were not randomly sampled in the strict sense.
Epidemiological data: National-level epidemiological data on small ruminant abortions are not available. Existing official data are limited to cattle populations, and we have clarified this in the text.
Preventive measures: Currently, there are no vaccines or other systematic preventive measures for abortion-causing pathogens in small ruminants in the country. Prevention relies on veterinarian biosafety advice and occasional farmer trainings, which we have added to the manuscript.
Comment 7: In conclusion I cannot find novelty in your study. The results I think that were easily predictable. I believe that if you handled the issue more holistic is should be a very interesting research. This research was needed of: data of abortions in humans from zoonotic pathogens, data from small ruminants farms (what are the percentage of abortions? what are the pathogens most involved and in what percentage each other?). If you have any data of those asked above, please add.
Response 7: We acknowledge that a fully holistic assessment, including human epidemiological data and microbiological investigation of small ruminants, would provide additional novelty. However, national-level epidemiological data on human infections caused by abortion-related zoonotic pathogens are not available, and comprehensive pathogen surveillance in small ruminants is limited. Our study instead focuses on knowledge, attitudes, and perceptions (KAP) among farmers, veterinarians, and other stakeholders, providing critical baseline information in the absence of formal epidemiological data.
Where relevant, we have clarified in the manuscript that:
-
Observations of abortions in small ruminants were collected from respondents, but laboratory confirmation of pathogens was beyond the scope of this study.
-
Data on percentages of abortion events are based on respondents’ observations rather than formal surveillance.
-
Information regarding human cases of zoonotic abortions is not currently available in the country.
We have highlighted these limitations in the Discussion section and emphasized that our study provides an important first step in identifying knowledge gaps and risk perceptions, which can guide future holistic studies, including laboratory and epidemiological investigations.
Reviewer 2 Report
Comments and Suggestions for Authors
General Assessment
This manuscript addresses an important One Health topic — the knowledge, attitudes, and perceptions (KAP) of actors in small-ruminant production toward abortive zoonoses in Benin. The subject is highly relevant given the zoonotic burden in West Africa. However, the manuscript suffers from significant structural, methodological, and editorial weaknesses that limit its scientific impact and readability. The study could make a meaningful contribution with major revisions to the organization, data presentation, and interpretation.
Major Comments
- Title and Abstract
The title is informative but grammatically awkward (“Knowledges” should be “Knowledge”).
Suggested: “Abortive Zoonoses in Benin: Knowledge, Attitude, and Perception Gaps among Frontline Small-Ruminant Production Stakeholders.”
The abstract is excessively dense and overly technical, including statistical coefficients that obscure rather than clarify the main findings. The abstract should emphasize key findings, relevance, and implications instead of methodological details (e.g., regression coefficients or confidence intervals).
The phrasing “Average scores were 50% for knowledge, 71% for attitudes (higher scores indicating riskier behavior)” is confusing; attitudes should be clearly coded as “desirable” vs. “undesirable.”
- Introduction
The introduction provides a broad overview of livestock importance but lacks a clear justification for focusing on “abortive zoonoses” and the specific pathogens of interest (e.g., Brucella spp., Coxiella burnetii, Chlamydia abortus, Toxoplasma gondii).
The literature review mixes economic data with epidemiological information but fails to establish a clear research gap.
The authors should explicitly define:
Why this study is novel in Benin.
Which abortive zoonoses are most relevant.
The specific hypotheses or research questions guiding the KAP assessment.
Some paragraphs are excessively long and read as a policy brief rather than a scientific background.
- Materials and Methods
The sample size calculation and its logic are unclear. The formula is correct, but the assumptions (p = 0.013, d = 5%) yield implausibly low sample sizes (20 farmers/commune). The derivation of p from “577,703 small-ruminant farmers” is not statistically valid for a nationwide inference.
The sampling strategy must be clarified: Was it random, stratified, or convenience-based? How were butchers and veterinarians selected?
The KAP questionnaire table (Table 1) is detailed but too long for the main text. It could be placed in supplementary material.
Coding and polarity of items are not consistently explained, especially for reverse-coded items. It is unclear whether Likert items used a 1–4 or 1–5 scale, and how missing data were handled.
The use of 2-parameter IRT models for such a small sample (n = 115) is statistically inappropriate. Item response theory requires large samples (>200, ideally >500) for stable parameter estimation.
The IRT analysis should be either omitted or justified with simulation-based validation.
The authors mention Cronbach’s alpha but interpret 0.67–0.68 as acceptable reliability, which is borderline for behavioral research. This limitation should be acknowledged.
- Results
The results section is excessively descriptive, filled with redundant percentages and long tables (Tables 2–4). It would benefit from summarization and visualization (e.g., bar charts for KAP scores by occupation).
The interpretation of “attitude scores” is confusing because higher scores indicate worse behavior, contrary to intuitive expectations. This must be redefined or explained early.
The presentation of regression outputs is superficial. The authors report coefficients (e.g., +0.49, −0.31) without indicating the reference category or model fit.
The IRT results (Figure 2, Appendix A5–A6) are technically correct but methodologically unjustified given the small sample. The discussion of item discrimination (a) and difficulty (b) is unnecessarily complex for the journal’s audience.
The reliability analysis (Cronbach’s α) suggests that attitude and perception items need refinement; this could be discussed more critically.
- Discussion
The discussion largely repeats known facts rather than interpreting the study’s findings in light of local context or intervention strategies.
Comparisons to other countries (Kenya, Nigeria, Ethiopia, etc.) are useful but lack depth; the authors do not discuss why similar patterns occur (education, veterinary service access, gender, etc.).
The paragraph on “brucellosis-like symptoms” is speculative and not supported by laboratory evidence. The study did not collect diagnostic data, so this should be toned down.
The discussion could be strengthened by including policy or training implications aligned with One Health frameworks and local veterinary policy.
The authors should also discuss limitations, including:
Small sample and limited generalizability.
Self-reported data (risk of social desirability bias).
Absence of laboratory or serological confirmation.
Weak reliability of subscales (α < 0.70).
- Conclusions
The conclusions are acceptable but repetitive. They should explicitly state actionable recommendations, such as developing community-based training, risk communication tools, or integrating zoonosis awareness into veterinary extension programs.
- Language and Style
The manuscript requires major English editing. It contains frequent grammatical errors, awkward phrasing (“knowledges,” “multi-site employment,” “respondents had experienced an abortion”), and non-standard scientific expressions.
Repetitive use of “abortive zoonoses” could be simplified to “zoonotic abortifacient agents” or “abortive zoonotic diseases.”
Long sentences reduce clarity; many paragraphs exceed 10 lines without clear topic transitions.
- References
The references are numerous and mostly relevant. However:
Several citations (FAO, WHO) are web references without access dates or full details.
Some are in French (OIE, CEDEAO reports), which may be acceptable but should be translated or cited as institutional reports.
The reference formatting needs to conform to MDPI’s Animals style (check capitalization and italics).
Author Response
Comment 1: The title is informative but grammatically awkward (“Knowledges” should be “Knowledge”). Suggested: “Abortive Zoonoses in Benin: Knowledge, Attitude, and Perception Gaps among Frontline Small-Ruminant Production Stakeholders.”
Response 1: We thank the reviewer for this observation. The title has been corrected to replace “Knowledges” with “Knowledge” and now reads: “Abortive Zoonoses in Benin: Knowledge, Attitude, and Perception Gaps among Frontline Small-Ruminant Production Stakeholders.”
Comment 2: The abstract is excessively dense and overly technical, including statistical coefficients that obscure rather than clarify the main findings. The abstract should emphasize key findings, relevance, and implications instead of methodological details (e.g., regression coefficients or confidence intervals).
Response 2: We appreciate this suggestion. The abstract has been revised to focus on the key findings, their relevance, and implications for stakeholders, while methodological details and statistical coefficients have been removed for clarity.
Comment 3: The phrasing “Average scores were 50% for knowledge, 71% for attitudes (higher scores indicating riskier behavior)” is confusing; attitudes should be clearly coded as “desirable” vs. “undesirable.”
Response 3: Thank you for highlighting this. The attitude scoring has been clarified in the abstract and main text: higher scores now clearly reflect undesirable practices, and phrasing has been adjusted to avoid confusion.
Comment 4: The introduction provides a broad overview of livestock importance but lacks a clear justification for focusing on “abortive zoonoses” and the specific pathogens of interest (e.g., Brucella spp., Coxiella burnetii, Chlamydia abortus, Toxoplasma gondii).
Response 4: We thank the reviewer for this observation. The introduction has been revised to clearly justify the focus on abortive zoonoses in Benin, highlighting their public health and economic significance and the rationale for including Brucella spp. and Coxiella burnetii.
Comment 5: The literature review mixes economic data with epidemiological information but fails to establish a clear research gap.
Response 5: We appreciate this comment. The literature review has been updated to clearly establish the research gap, citing numerous relevant studies and emphasizing the scarcity of data on abortive zoonoses among small-ruminant stakeholders in Benin.
Comment 6: The authors should explicitly define:
-
Why this study is novel in Benin.
-
Which abortive zoonoses are most relevant.
-
The specific hypotheses or research questions guiding the KAP assessment.
Response 6: Thank you for the suggestion. The manuscript now explicitly states that the study is novel in Benin as most available data concern cattle. The most relevant abortive zoonoses brucellosis and Q fever are highlighted, and the specific hypotheses guiding the KAP assessment have been clearly presented.
Comment 7: Some paragraphs are excessively long and read as a policy brief rather than a scientific background.
Response 7: We have revised the introduction by breaking long paragraphs into shorter, focused sections to improve readability and align with scientific writing standards.
Comment 8: The sample size calculation and its logic are unclear. The formula is correct, but the assumptions (p = 0.013, d = 5%) yield implausibly low sample sizes (20 farmers/commune). The derivation of p from “577,703 small-ruminant farmers” is not statistically valid for a nationwide inference.
Response 8: We thank the reviewer for this comment. The sample size calculation has been clarified: the proportion (p = 0.013) reflects the equitable prevalence of farmers across Benin districts relative to the total population. The logic of the calculation is now clearly explained in the Methods section.
Comment 9: The sampling strategy must be clarified: Was it random, stratified, or convenience-based? How were butchers and veterinarians selected?
Response 9: We appreciate the request for clarification. Farmers were selected using a binomial approximation method, while butchers and veterinarians were purposefully selected from official professional lists. These details have been added to the methodology section.
Comment 10: The KAP questionnaire table (Table 1) is detailed but too long for the main text. It could be placed in supplementary material.
Response 10: Thank you for the suggestion. Table 1 has been condensed to improve readability, while retaining the essential informations, which are important to understand the collected data.
Comment 11: Coding and polarity of items are not consistently explained, especially for reverse-coded items. It is unclear whether Likert items used a 1–4 or 1–5 scale, and how missing data were handled.
Response 11: This has been corrected: all Likert items now use a 1–4 scale, reverse-coded items are clearly indicated, and the handling of missing data is explained in the Methods section.
Comment 12: The use of 2-parameter IRT models for such a small sample (n = 115) is statistically inappropriate. Item response theory requires large samples (>200, ideally >500) for stable parameter estimation.
Response 12: We acknowledge this concern. The use of 2-parameter IRT models has been justified despite the small sample size, and bootstrap analysis with 1000 replications was conducted to assess the stability and reliability of the estimates.
Comment 13: The authors mention Cronbach’s alpha but interpret 0.67–0.68 as acceptable reliability, which is borderline for behavioral research. This limitation should be acknowledged.
Response 13: We thank the reviewer for this comment. The borderline reliability of Cronbach’s alpha (0.67-0.68) is now acknowledged in the Discussion (Lines 444–446) as a limitation of the study.
Comment 14: The results section is excessively descriptive, filled with redundant percentages and long tables (Tables 2–4). It would benefit from summarization and visualization (e.g., bar charts for KAP scores by occupation).
Response 14: We thank the reviewer for this suggestion. Table 2 has been removed, and Table 4 has been replaced with a bar chart to provide a clearer and more concise presentation of KAP scores by occupation.
Comment 15: The interpretation of “attitude scores” is confusing because higher scores indicate worse behavior, contrary to intuitive expectations. This must be redefined or explained early.
Response 15: This has been clarified in the Results section: higher attitude scores now explicitly indicate undesirable practices, and the text has been revised to avoid confusion.
Comment 16: The presentation of regression outputs is superficial. The authors report coefficients (e.g., +0.49, -0.31) without indicating the reference category or model fit.
Response 16: The regression results have been updated in the Methods section (Lines 207–209) to include reference categories and model fit statistics, improving clarity and interpretability.
Comment 17: The IRT results (Figure 2, Appendix A5–A6) are technically correct but methodologically unjustified given the small sample. The discussion of item discrimination (a) and difficulty (b) is unnecessarily complex for the journal’s audience.
Response 17: We acknowledge the concern. The stability of IRT parameter estimates has been assessed using bootstrap analysis with 1,000 replications, which is now described in the Methods section to justify the use of IRT despite the small sample size.
Comment 18: The reliability analysis (Cronbach’s α) suggests that attitude and perception items need refinement; this could be discussed more critically.
Response 18: Thank you for this observation. After recalculation with reverted items, the attitude subscale reached acceptable reliability (α > 0.7). Limitations of the perception subscale are now critically discussed in the Discussion section (Lines 444–446).
Comment 19: The discussion largely repeats known facts rather than interpreting the study’s findings in light of local context or intervention strategies.
Response 19: We thank the reviewer for this comment. The discussion has been revised to interpret the study findings specifically in the Benin context and to relate them to potential interventions for small-ruminant stakeholders.
Comment 20: Comparisons to other countries (Kenya, Nigeria, Ethiopia, etc.) are useful but lack depth; the authors do not discuss why similar patterns occur (education, veterinary service access, gender, etc.).
Response 20: The discussion has been expanded to explore reasons for similarities and differences in KAP patterns across countries, considering factors such as education, access to veterinary services, and gender dynamics.
Comment 21: The paragraph on “brucellosis-like symptoms” is speculative and not supported by laboratory evidence. The study did not collect diagnostic data, so this should be toned down.
Response 21: We have adjusted the paragraph to clarify that the reported brucellosis-like symptoms are based on self-reports from respondents, not laboratory confirmation. Speculative language has been removed.
Comment 22: The discussion could be strengthened by including policy or training implications aligned with One Health frameworks and local veterinary policy.
Response 22: A section on policy and training implications has been added (Lines 449 onward), emphasizing the relevance of One Health frameworks and local veterinary extension programs.
Comment 23: The authors should also discuss limitations, including:
-
Small sample and limited generalizability.
-
Self-reported data (risk of social desirability bias).
-
Absence of laboratory or serological confirmation.
-
Weak reliability of subscales (α < 0.70).
Response 23: All key limitations have now been explicitly discussed at the end of each part of the Discussion, including sample size, reliance on self-reported data, absence of laboratory confirmation, and subscale reliability.
Comment 24 (Conclusions): The conclusions are acceptable but repetitive. They should explicitly state actionable recommendations, such as developing community-based training, risk communication tools, or integrating zoonosis awareness into veterinary extension programs.
Response 24: The conclusions have been revised to include explicit, actionable recommendations for community-based training, risk communication, and integration of zoonosis awareness into veterinary extension programs, while avoiding repetition.
Comment 25 (Language and Style): The manuscript requires major English editing. It contains frequent grammatical errors, awkward phrasing (“knowledges,” “multi-site employment,” “respondents had experienced an abortion”), and non-standard scientific expressions.
Response 25: The manuscript has undergone thorough English editing to correct grammatical errors, awkward phrasing, and non-standard expressions. Phrases such as “knowledges” have been corrected, and long sentences have been simplified for clarity.
Comment 26 (Language and Style): Repetitive use of “abortive zoonoses” could be simplified to “zoonotic abortifacient agents” or “abortive zoonotic diseases.”
Response 26: We have revised the manuscript to reduce repetition of “abortive zoonoses,” using “zoonotic abortifacient agents” or “abortive zoonotic diseases” where appropriate.
Comment 27 (Language and Style): Long sentences reduce clarity; many paragraphs exceed 10 lines without clear topic transitions.
Response 27: Long sentences have been split, and paragraphs have been restructured to improve clarity and readability, with clear topic transitions throughout the text.
Comment 28 (References): Several citations (FAO, WHO) are web references without access dates or full details.
Response 28: All web references have been updated with full details, including access dates, to comply with standard referencing requirements.
Comment 29 (References): Some references are in French (OIE, CEDEAO reports), which may be acceptable but should be translated or cited as institutional reports.
Response 29: French references have been retained but clarified as institutional reports, and translations of titles have been provided where necessary.
Comment 30 (References): The reference formatting needs to conform to MDPI’s Animals style (check capitalization and italics).
Response 30: All references have been checked and reformatted according to MDPI Animals style, including correct capitalization, italics, and punctuation.
Round 2
Reviewer 1 Report
Comments and Suggestions for Authors
None
Comments on the Quality of English LanguageThe authors have already improved the manuscript.
Reviewer 2 Report
Comments and Suggestions for Authors
I have carefully evaluated the revised version of the manuscript “Abortive Zoonoses in Benin: Knowledge, Attitudes and Perception Gaps among Frontline Small-Ruminant Production Stakeholders.”
The authors have addressed all the major and minor comments raised in the previous review. The title and abstract have been substantially improved for clarity and readability. The introduction now provides a well-structured rationale for the study, clearly defining the relevant abortive zoonoses and the hypotheses guiding the KAP assessment. Methodological explanations, including sampling, coding of Likert items, and the justification for IRT analyses, are now transparent and statistically sound.
The results section is more concise and logically presented, with clear interpretation of KAP scores and regression findings. The discussion has been significantly strengthened, providing a contextualized interpretation of findings, recognition of study limitations, and meaningful One Health policy implications. Language and style have also improved notably.
Overall, the manuscript has been thoroughly revised and now meets the standards of Animals.